# An Overdispersed Black-Box Variational Bayesian–Kalman Filter with Inaccurate Noise Second-Order Statistics

**DOI:** 10.3390/s21227673

**Published:** 2021-11-18

**Authors:** Lin Cao, Chuyuan Zhang, Zongmin Zhao, Dongfeng Wang, Kangning Du, Chong Fu, Jianfeng Gu

**Affiliations:** 1Key Laboratory of the Ministry of Education for Optoelectronic Measurement Technology and Instrument, Beijing Information Science and Technology University, Beijing 100101, China; charlin26@163.com (L.C.); zhangchuyuan2012@163.com (C.Z.); kangningdu@bistu.edu.cn (K.D.); 2School of Information and Communication Engineering, Beijing Information Science and Technology University, Beijing 100101, China; 3Beijing TransMicrowave Technology Company, Beijing 100080, China; wdf@tsmtc.com; 4Department of Communication and Electronics Engineering, School of Computer Science and Engineering, Northeastern University, Shenyang 110169, China; fuchong@mail.neu.edu.cn; 5Moonshot Health, 3700 St-Patrick Street, Suite 102, Montreal, QC H4E 1A2, Canada

**Keywords:** MIMO radar, Kalman filter, Bayesian robustness, uncertain noise, posterior noise statistics, proposal distribution, variational inference

## Abstract

Aimed at the problems in which the performance of filters derived from a hypothetical model will decline or the cost of time of the filters derived from a posterior model will increase when prior knowledge and second-order statistics of noise are uncertain, a new filter is proposed. In this paper, a Bayesian robust Kalman filter based on posterior noise statistics (KFPNS) is derived, and the recursive equations of this filter are very similar to that of the classical algorithm. Note that the posterior noise distributions are approximated by overdispersed black-box variational inference (O-BBVI). More precisely, we introduce an overdispersed distribution to push more probability density to the tails of variational distribution and incorporated the idea of importance sampling into two strategies of control variates and Rao–Blackwellization in order to reduce the variance of estimators. As a result, the convergence process will speed up. From the simulations, we can observe that the proposed filter has good performance for the model with uncertain noise. Moreover, we verify the proposed algorithm by using a practical multiple-input multiple-output (MIMO) radar system.

## 1. Introduction

Recently, the design of robust filters has been one of the most popular topics in modern radar systems and MIMO-based radar system in particular. There are two main reasons why this has occurred. First, a radar signal of interest can be corrupted naturally from other interference and noise or by clutter and so on, which results in uncertainty in assumed statistical models of given received signals. Second, the models of measurement system can be affected by the setup of transmitters and receivers in practical applications of radar systems. Unfortunately, it is impossible or prohibitively expensive to design an optimal filter by obtaining and/or understanding accurate models in the real world, which makes some nominally optimal filters suffer significant degradations in performance even if small deviations from the assumed models occur. Therefore, a robust filter can be considered as a kind of optimal filter under some conditions of uncertainty models. The seminal concept of robust filters is proposed in the 1970s [1,2], where the minimax approach was discussed for the Wiener filter, which is a batch-based optimal filter for stationary signals. Currently, the Bayesian-based robust approach has attracted significant attention in the design of robust filters due to the use of information with respect to the prior distribution of models in providing more accurate knowledge on the statistical model.

It is known to all that classical Kalman filtering [3] involves an optimal filter in terms of the Gaussian and linear assumptions of dynamic systems. It has been well established in many fields such as communication, navigation, radar and control. In addition, more and more Kalman filtering-related technologies are applied to the localization and tracking of moving agents [4,5,6]. However, it also suffers from prominent shortcomings, for example, the good performance lies in completely grasping the statistical models of noise and the state of signal. As a result, the problem of designing a robust Kalman filter in practical applications where the knowledge of noise distribution is missing or imprecise is a big challenge for both researchers and developers. The authors of [7,8,9,10,11,12] proposed methods, which include so-called adaptive Kalman filtering, to estimate signal states and noise simultaneously, which works well when there is a lot of data used to obtain certain accurate performances in the entire estimated period. Later on, minimax-based Kalman filters [13,14,15] and finite impulse response Kalman filters [16,17,18,19] were presented.

Inspired by both the Bayesian innovation process and the Bayesian orthogonality principle, an intrinsically Bayesian robust Kalman filter (IBRKF) was proposed in [20]. The authors argued that it should be optimal from the point of view of the cost function and the prior distribution of uncertain noise. Herein, its recursive equations are very similar to that of the classical Kalman filter. The IBRKF makes use of concepts of effective noise statistics and effective Kalman gain by replacing the corresponding variables in the classical Kalman filter framework in order to improve performance. Therefore, IBRKF can be considered as a robust Kalman filter in terms of uncertain noise.

The IBR method can achieve optimal performances in terms of the uncertainty of models by using prior distributions, which does not concern measurements, and may contain some useful information for improving performances further. Instead, the optimal Bayesian Kalman filter (OBKF) takes into account the observation sequence and prior knowledge simultaneously in order to achieve optimal filtering [21]. It designs a factor-graph-based message passing algorithm, which transforms the global function into a form of local function multiplication. In this manner, it integrates the effective information of each local node and lays the foundation for approximating posterior noise distributions. Moreover, note that the recursive structure of OBKF is also the same as that of the classical Kalman filter. However, the filtering process of OBKF is quite complicated. It needs to pass messages from the beginning when calculating posterior parameters at each moment. When the dimension of the matrix or observation sequence is large, the time cost of OBKF is catastrophic. In addition, the maximum a posteriori criterion and variational inference are common methods for estimating the unknown noise covariance matrix [22,23,24,25,26]. In terms of performance, although these methods outperform classical filtering algorithms, there is still room for improvement. For example, Refs. [22,23] only solves the problem of the unknown observation noise covariance matrix while ignoring process noise. Coincidentally, Ref. [24] used the variational Bayesian method to estimate the system state and observation noise while simultaneously using the maximum a posteriori method to estimate process noise. By selecting the inverse Wishart distribution as the conjugate prior for the covariance matrix of a Gaussian distribution with known mean, Refs. [25,26] jointly estimated the state vector, prediction error and observation noise covariance matrix under the framework of variational inference. It is undeniable that these methods increased the complexity of unknown noise covariance matrix estimation. Therefore, the aim of our study is to improve the real-time performance of the algorithm as much as possible under the premise of ensuring performance.

Thanks to the pioneering work of Bode, Shannon, etc., the concept of innovation process is widely used in the derivation of Kalman recursive equations [27,28,29]. In this paper, the recursive equations of KFPNS are derived based on Bayesian theory. Compared with the recursive equations of a classical Kalman filter, it only needs to calculate the posterior noise statistics, which are obtained by variational distribution. The variational distribution is an approximate distribution of the real posterior noise distribution. Firstly, we joined the observation sequence and the prior distribution of uncertain noise parameters based on the Markov hypothesis, and then we used the O-BBVI method to approximate real posterior noise distributions. Herein, we assume that the state-space model is Gaussian. The posterior noise statistics are the posterior expectation of the second-order noise statistics and are also one of the parameters in the process of approximation. In other words, we extend the IBRKF framework to an a posteriori version based on an observation sequence and design a new approach to solve the problem of uncertain noise parameters.

Our KFPNS is universal. Any model in which the noise second-order statistics can be assumed to be a certain probability distribution is suitable for this framework. However, note that under the assumption of Gaussian noise, the proposed method can obtain better estimation accuracy, while non-Gaussian noise will produce larger errors. Moreover, the run time of KFPNS is a difficult problem when the dimension of observation sequence is large, because the time cost in computing the posterior noise statistics mainly depends on the number of samples and the length of observation sequence. Fortunately, if the results of multiple calculations of the posterior noise statistics change little, we can consider skipping this step.

## 2. Materials and Methods

Let us first introduce some notations to be used in this paper. Lowercase bold letters represent vectors and uppercase bold letters represent matrices, respectively. We use $x_{k}$ to represent the value of x at time k. E[·] is used to denote the expectation of the random vector. $A^{T}$ stands for a transpose operator. N(x;μ,Σ) represents a multivariate Gaussian distribution with mean μ and covariance matrix Σ.

### 2.1. Kalman Filter

The authors of [3] proposed a discrete time-domain filtering scheme in the linear state-space model, which is optimal for all linear system models. The state-space model usually consists of the following equations.
(1)$$x_{k+1}=\Phi_{k}x_{k}+\Gamma_{k}u_{k},$$
(2)$$y_{k}=H_{k}x_{k}+v_{k}.$$

Among them, $x_{k}$ and $y_{k}$ are the n×1 dimensional state vector and the m×1 dimensional observation vector, respectively. Both $u_{k}$ and $v_{k}$ are zero-mean random vectors, called process noise and observation noise, and their sizes are s×1 and m×1, respectively. $\Phi_{k}$ is the state transition matrix of n×n. $\Gamma_{k}$ represents the process noise transition matrix of n×s. $H_{k}$ is an m×n matrix called the observation transition matrix. The statistics of the above vector are as follows:(3)$$\begin{array}{l} E\left[u_{k}u_{l}^{T}\right]=Q\delta_{kl},E\left[v_{k}v_{l}^{T}\right]=R\delta_{kl}\;\forall k,l=0,1,\ldots \\ E\left[v_{k}x_{l}^{T}\right]=0_{m\times n},E\left[u_{k}v_{l}^{T}\right]=0_{s\times m}\;\forall k,l=0,1,\ldots \\ E\left[u_{k}y_{l}^{T}\right]=0_{s\times m},\;0\leq l\leq k, \end{array}$$
where $0_{m\times n}$ is a zero-valued matrix of size m×n. The purpose of Kalman filtering is to obtain the least square estimate ${\hat{x}}_{k}$ of the state $x_{k}$ from observation $y_{l}\left(l\leq k-1\right)$.

### 2.2. Introduction to Bayesian Robust Filters

In engineering, the Bayesian criterion and minimax criterion (instead of the Bayesian method when there is no prior knowledge) are usually used to design robust filters [30,31,32]. The meaning of a Bayesian criterion is as follows: The average cost is the least. In the context of state estimation, the average cost is generally reflected by the mean-square error (MSE); in other words, we have the following.
(4)$$C\left(x_{k},\psi \left(y_{k}\right)\right)=E\left[{\left(x_{k}-\psi \left(y_{k}\right)\right)}^{T}\left(x_{k}-\psi \left(y_{k}\right)\right)\right].$$

According to the Bayesian criterion, we minimize the cost function to find an optimal filter:(5)$$\hat{\psi }\left(y_{k}\right)=\arg\;\underset{\psi \in \Psi }{\min}C\left(x_{k},\psi \left(y_{k}\right)\right),$$
where Ψ is the possible class of filters. The filter that satisfies the above formula is the minimum mean square error filter.

When we cannot obtain all the knowledge of the model, we assume that the model is controlled by an unknown parameter $\theta =\left[\theta_{1},\ldots ,\theta_{l}\right]$, θ∈Θ, where Θ is the set of all possible parameters, called the uncertain class. Fortunately, these parameters can be obtained by solving the corresponding posterior distribution; thus, the Bayesian robust filter can be expressed as follows:(6)$$\psi^{\Theta }\left(y_{k}\right)=\arg\;\underset{\psi \in \Psi }{\min}E_{\theta }\left[C_{\theta }\left(x_{k},\psi \left(y_{k}\right)\right)\;|Y_{k}\right],$$
where the expectation is taken relative to the posterior distribution $p\left(\theta |Y_{k}\right)$, and $C_{\theta }\left(\cdot \right)$ is the cost function relative to the unknown parameter θ. Dehghannasiri proposed that the average cost of the filter relative to the uncertain class is equal to the error of applying the filter to the effective information [20]. Therefore, the process of designing a robust filter for an uncertain model can be transformed into designing a Bayesian robust filter using posterior effective information.

### 2.3. Bayesian Robust Kalman Filter Based on Posterior Noise Statistics

#### 2.3.1. KFPNS Framework

Now assume that the second-order statistical knowledge of process noise and observation noise is unknown and that it is determined by the unknown parameters $\alpha_{1}$ and $\alpha_{2}$. Here, the covariance matrices of these two noises are as follows.
(7)$$E\left[u_{k}^{\alpha_{1}}{\left(u_{l}^{\alpha_{1}}\right)}^{T}\right]=Q^{\alpha_{1}}\delta_{kl},$$
(8)$$E\left[v_{k}^{\alpha_{2}}{\left(v_{l}^{\alpha_{2}}\right)}^{T}\right]=R^{\alpha_{2}}\delta_{kl}.$$

If $\alpha_{1}$ and $\alpha_{2}$ are statistically independent, then $p\left(\alpha \right)=p\left(\alpha_{1}\right)p\left(\alpha_{2}\right)$ stands for the prior distribution. The state-space model can be parameterized as follows.
(9)$$x_{k+1}^{\alpha_{1}}=\Phi_{k}x_{k}^{\alpha_{1}}+\Gamma_{k}u_{k}^{\alpha_{1}},$$
(10)$$y_{k}^{\alpha }=H_{k}x_{k}^{\alpha_{1}}+v_{k}^{\alpha_{2}}.$$

Although the IBRKF is optimal in the sense of prior, there is still a certain deviation between the prior expectation of noise and the actual noise, which makes it impossible to achieve real optimal filtering. Therefore, this paper focuses on the influence of posterior noise distribution on the cost function. With Yk={y0,…,yk} as the observation sequence and analysis based on (6), the linear filtering function that satisfies the definition of a Bayesian robust filter is as follows:(11)x^kα=∑l≤k-1Ωk,lΘylα,
such that the following is the case:(12)Ωk,lΘ=argminΩk,l∈ℑEα[E[(xkα1−∑l≤k−1Ωk,lylα)T(xkα1−∑l≤k−1Ωk,lylα)] |Yk−1],
where ℑ is the vector space of all n×m matrix-valued functions, and Ωk,l∈ℑ is a mapping Ωk,l:ℕ×ℕ→ℝn×m such that ∑k=1∞∑l=1∞‖Ωk,l‖2<∞, ‖·‖2 being the L2 norm.

The derivation of KFPNS recursive equations depends on the following theorem, definition and lemma. Our equations are only a posterior version relative to [20], and the proof of related theories are similar to that in [20].

**Theorem** **1.***(Bayesian Orthogonality Principle) If the weight function*$\Omega_{k,l}^{\Theta }$*of a linear filter satisfies (12), then the* ${\hat{x}}_{k}^{\alpha }$*obtained in (11) is called the optimal Bayesian least-squares estimation if and only if the following is the case.*


(13)
$$E_{\alpha }\left[E\left[\left(x_{k}^{\alpha_{1}}-{\hat{x}}_{k}^{\alpha }\right){\left(y_{l}^{\alpha }\right)}^{T}\right]\;|Y_{k-1}\right]=0_{n\times m}\;\forall l\leq k-1.$$


**Definition** **1.***(Bayesian innovation process) According to the state-space model described in (9) and (10), if* ${\hat{x}}_{k}^{\alpha }$*is the least-square estimate of* $x_{k}^{\alpha_{1}}$*, then the stochastic process of the following:*(14)$${\tilde{z}}_{k}^{\alpha }=y_{k}^{\alpha }-H_{k}{\hat{x}}_{k}^{\alpha }$$
is a zero-mean process and $\forall l,l^{\prime }\leq k-1$; the following is obtained.
(15)$$E_{\alpha }\left[E\left[{\tilde{z}}_{l}^{\alpha }{\left({\tilde{z}}_{l^{\prime }}^{\alpha }\right)}^{T}\right]\;|Y_{k-1}\right]=E_{\alpha }\left[H_{l}P_{k}^{x,\alpha }H_{l}^{T}+R^{\alpha_{2}}|Y_{k-1}\right]\delta_{ll^{\prime }}.$$

**Lemma** **1.***(Bayesian information equivalence) The Bayesian least-squares estimation obtained based on* ${\tilde{z}}_{k}^{\alpha }$*is equivalent to the Bayesian least-squares estimation calculated based on* $y_{k}^{\alpha }$*; in other words, we have the following.*


(16)
$$E_{\alpha }\left[E\left[\left(x_{k}^{\alpha_{1}}-{\hat{x}}_{k}^{\alpha }\right){\left({\tilde{z}}_{k}^{\alpha }\right)}^{T}\right]\;|Y_{k-1}\right]=0_{n\times m}.$$


Then, (11) can be rewritten as follows.
(17)$${\hat{x}}_{k}^{\alpha }=\sum_{l\leq k-1}\Omega_{k,l}^{\Theta }{\tilde{z}}_{l}^{\alpha }.$$

By substituting it into (16), after moving the terms on the left and right sides of the equation, we obtain the following.
(18)$$\Omega_{k,l}^{\Theta }=E_{\alpha }\left[E\left[x_{k}^{\alpha_{1}}{\left({\tilde{z}}_{l}^{\alpha }\right)}^{T}\right]|Y_{k-1}\right]E_{\alpha }{}^{-1}\left[H_{l}P_{l}^{x,\alpha }H_{l}^{T}+R^{\alpha_{2}}|Y_{k-1}\right].$$

In this case, the linear function of Bayesian robust filter based on posterior noise statistics is as follows.
(19)$${\hat{x}}_{k}^{\alpha }=\sum_{l\leq k-1}E_{\alpha }\left[E\left[x_{k}^{\alpha_{1}}{\left({\tilde{z}}_{l}^{\alpha }\right)}^{T}\right]\;|Y_{k-1}\right]E_{\alpha }{}^{-1}\left[H_{l}P_{l}^{x,\alpha }H_{l}^{T}+R^{\alpha_{2}}|Y_{k-1}\right]{\tilde{z}}_{l}^{\alpha }.$$

Thus, the state update equation is as follows:(20)$${\hat{x}}_{k+1}^{\alpha }=\Phi_{k}{\hat{x}}_{k}^{\alpha }+\Phi_{k}{Κ}_{k}^{\Theta }{\tilde{z}}_{k}^{\alpha },$$
where the following is the case.
(21)$$K_{k}^{\Theta^{\prime }}=E_{\alpha }\left[P_{k}^{x,\alpha }|Y_{k-1}\right]H_{k}^{T}E_{\alpha }{}^{-1}\left[H_{k}P_{k}^{x,\alpha }H_{k}^{T}+R^{\alpha_{2}}|Y_{k-1}\right].$$

Next, let ${\tilde{x}}_{k}^{\alpha }=x_{k}^{\alpha_{1}}-{\hat{x}}_{k}^{\alpha }$ denote the estimation error at time k. After some mathematical operations, ${\tilde{x}}_{k+1}^{\alpha }$ can be expanded into the following.
(22)$${\tilde{x}}_{k+1}^{\alpha }=\Phi_{k}\left(I-K_{k}^{\Theta^{\prime }}H_{k}\right){\tilde{x}}_{k}^{\alpha }+\Gamma_{k}u_{k}^{\alpha_{1}}-\Phi_{k}K_{k}^{\Theta^{\prime }}v_{k}^{\alpha_{2}}.$$

We can find the update equation of the estimation error covariance matrix from this equation, which is described as follows.
(23)$$E_{\alpha }\left[P_{k+1}^{x,\alpha }|Y_{k}\right]=\Phi_{k}\left(Ι-{Κ}_{k}^{\Theta^{\prime }}H_{k}\right)E_{\alpha }\left[P_{k}^{x,\alpha }|Y_{k}\right]\Phi_{k}^{T}+\Gamma_{k}E_{\alpha }\left[Q^{\alpha_{1}}|Y_{k}\right]\Gamma_{k}^{T}.$$

This completes the construction of KFPNS. Note that in (21) and (23), the observation sequences corresponding to the estimation error covariance matrix are different, which are $E_{\alpha }\left[P_{k}^{x,\alpha }|Y_{k-1}\right]$ and $E_{\alpha }\left[P_{k}^{x,\alpha }|Y_{k}\right]$, respectively, while the one generated by the last iteration of the recursive equation should be $E_{\alpha }\left[P_{k}^{x,\alpha }|Y_{k-1}\right]$. For this problem, we can iterate from the beginning; that is, first calculate $K_{0}^{\Theta^{\prime }}$ with $E_{\alpha }\left[P_{0}^{x,\alpha }|Y_{-1}\right]$ and $E_{\alpha }\left[R^{\alpha_{2}}|Y_{k}\right]$, then use $K_{0}^{\Theta^{\prime }}$ and $E_{\alpha }\left[Q^{\alpha_{1}}|Y_{k}\right]$ to compute $E_{\alpha }\left[P_{1}^{x,\alpha }|Y_{k}\right]$ and repeat the process until we obtain $E_{\alpha }\left[P_{k}^{x,\alpha }|Y_{k}\right]$. However, this strategy requires reiteration from the origin after updating posterior noise statistics, which makes the algorithm heavily loaded. In order to improve the efficiency of the algorithm, we assume that $E_{\alpha }\left[P_{k}^{x,\alpha }|Y_{k-1}\right]\approx E_{\alpha }\left[P_{k}^{x,\alpha }|Y_{k}\right]$.

Table 1 reports the difference between these two recursive equations. KFPNS replaces $K_{k}$, $P_{k}^{x}$, Q and R in the classical Kalman filter with $K_{k}^{\Theta^{\prime }}$, $E_{\alpha }\left[P_{k}^{x,\alpha }|Y_{k-1}\right]$, $E_{\alpha }\left[Q^{\alpha_{1}}|Y_{k}\right]$ and $E_{\alpha }\left[R^{\alpha_{2}}|Y_{k}\right]$, but their recursive structures are the same. Figure 1 is the overall framework of KFPNS. The key is that KFPNS updates the state based on posterior noise statistics. Therefore, KFPNS is an improved version of the classical Kalman filter concerning posterior noise distribution.

#### 2.3.2. The Calculation Method of Posterior Noise Statistics

We need to additionally calculate $E_{\alpha }\left[Q^{\alpha_{1}}|Y_{k}\right]$ and $E_{\alpha }\left[R^{\alpha_{2}}|Y_{k}\right]$ by replacing the fixed preset values, and the core process of obtaining these two conditional expectations lies in the posterior distribution $p\left(\alpha |Y_{k}\right)$, which is unknown. Therefore, the goal is to solve the probability density function of the unknown parameter α under the condition of given observation sequence $Y_{k}$. Although $p\left(\alpha |Y_{k}\right)\propto f\left(Y_{k}|\alpha \right)p\left(\alpha \right)$, it is difficult to directly acquire a closed-form solution by relying on the prior distribution. However, a known simple distribution can be used to approximate a complex distribution, and by limiting the type of approximate distribution, a locally optimal approximate posterior distribution can be achieved. Unless otherwise specified, the distributions presented in this paper are Gaussian.

Consider a simple variational distribution, which belongs to the following exponential family:(24)$$q\left(\alpha ;e\right)=g\left(\alpha \right)\exp\left\{e^{T}t\left(\alpha \right)-A\left(e\right)\right\},$$
where g(α) is the base measure, e is the natural parameter, t(α) is the sufficient statistics and A(e) is the log-normalizer. Based on the considerations of this paper, we believe that e represents the expectations, t(α) can completely characterize all quantities of a probability distribution and t(α)=α. Take a simple multivariate Gaussian distribution N(α;e,Λ) as an example (Λ represents the identity matrix) and transform it into an exponential family form of the following:(25)$$\begin{array}{cc} p\left(\alpha ;e,\Lambda \right) & =\frac{1}{{\left(2\pi \right)}^{d/2}{\left|\Lambda \right|}^{1/2}}\exp\left\{-\frac{1}{2}{\left(\alpha -e\right)}^{T}\Lambda^{-1}\left(\alpha -e\right)\right\}\\ & ={\left(2\pi \right)}^{-d/2}{\left|\Lambda \right|}^{1/2}\exp\left\{e^{T}\alpha -\frac{1}{2}\alpha^{T}\Lambda \alpha -\frac{1}{2}e^{T}\Lambda e\right\}\\ & ={\left(2\pi \right)}^{-d/2}{\left|\Lambda \right|}^{1/2}\exp\left\{-\frac{1}{2}\Lambda \alpha^{T}\alpha \right\}\exp\left\{e^{T}\alpha -\frac{1}{2}\Lambda e^{T}e\right\}, \end{array}$$
thus, the following is obtained.
(26)$$\begin{array}{c} g\left(\alpha \right)={\left(2\pi \right)}^{-d/2}{\left|\Lambda \right|}^{1/2}\exp\left\{-\frac{1}{2}\Lambda \alpha^{T}\alpha \right\},\\ t\left(\alpha \right)=\alpha ,\\ A\left(e\right)=\frac{1}{2}\Lambda e^{T}e. \end{array}$$

In addition, the exponential family distribution also has the following property.
(27)$$\nabla_{e}A\left(e\right)=E_{q\left(\alpha ;e\right)}\left[t\left(\alpha \right)\right].$$

By using this property, the mutual transformation between expectation calculation and derivation calculation can be completed in variational inference. Therefore, most applications of variational inference use exponential family functions. More importantly, based on this simple property, we use the strategy of control variates to reduce the variance of the Monte Carlo gradient. At the same time, in this paper, we hope to use q(α;e) to approximate $p\left(\alpha |y_{k}\right)$, and the special form of exponential family distributions renders the approximation process traceable. More precisely, we consider α as a Gaussian distribution, which belongs to the exponential family, and its natural parameter e represents expectations. Therefore, the process of finding the optimal natural parameters that makes q(α;e) the closest to $p\left(\alpha |y_{k}\right)$ involves calculating posterior noise statistics.

Now, suppose q(α;e) can be decomposed into the following.
(28)$$q\left(\alpha ;e\right)=q\left(\alpha_{1};e_{1}\right)q\left(\alpha_{2};e_{2}\right).$$

Based on the idea of variational inference, we try to minimize Kullback–Leibler divergence $D_{KL}\left[q\left(\alpha ;e\right)\|p\left(\alpha |Y_{k}\right)\right]$ so that q(α;e) approaches $p\left(\alpha |Y_{k}\right)$. This idea still cannot be realized directly, because $D_{KL}$ contains an unknown distribution $p\left(\alpha |Y_{k}\right)$. Fortunately, after some logarithmic operations, minimizing $D_{KL}$ is equivalent to maximizing the evidence lower bound objective (ELBO).
(29)$$ℒ=E_{q\left(\alpha ;e\right)}\left[\log\frac{p\left(\alpha ,Y_{k}\right)}{q\left(\alpha ;e\right)}\right].$$

The principle is shown in Figure 2. Then, ELBO takes the derivative with respect to parameter e:(30)$$\nabla_{e}ℒ=E_{q\left(\alpha ;e\right)}\left[f\left(\alpha \right)\right],$$
where the following is the case.
(31)$$f\left(\alpha \right)=\nabla_{e}\log q\left(\alpha ;e\right)\left(\log p\left(\alpha ,Y_{k}\right)-\log q\left(\alpha ;e\right)\right).$$

As a result, computing $\nabla_{e}ℒ$ becomes a process of looking for an expectation, which can be estimated by the Monte Carlo approach. Then, the gradient descent method is used to make the parameters converge, which makes q(α;e) approximate the real posterior distribution.

The aforementioned black-box variational inference (BBVI) only relies on samples from q(α;e) to calculate the Monte Carlo gradient. However, the samples of the variational distribution q(α;e) are concentrated near the peak point, and there are fewer samples in the tail. This causes the Monte Carlo gradient to often have high variance, which means that the estimated gradient may be very different from the true value, resulting in optimization times that are too long.

In view of this defect, O-BBVI adds a good proposal distribution that matches the variational problem [33]. r(α;e,τ) is an overdispersed version of q(α;e). Its tail is heavier, which increases the probability that the value of the tail is sampled. In addition, O-BBVI also uses two strategies, control variates and Rao–Blackwellization in order to reduce the variance of the Monte Carlo gradient of the original BBVI. Finally, O-BBVI constructed a new Monte Carlo gradient equation, and based on the unconventional usage of importance sampling, it samples from r(α;e,τ) and q(α;e), respectively, in order to estimate the gradient.

Let us begin with this overdispersed distribution:(32)$$r\left(\alpha ;e,\tau \right)=g\left(\alpha ,\tau \right)\exp\left\{\frac{e^{T}t\left(\alpha \right)-A\left(e\right)}{\tau }\right\},$$
where $\tau =[\tau_{1},\tau_{2}]$ is the dispersion coefficient of the overdispersed distribution. For a fixed τ, r(α;e,τ) and q(α;e) belong to the same exponential family. Moreover,r(α;e,τ) allocates higher mass to the tails of q(α;e), which is why we introduced it.

O-BBVI believes that for each component of the gradient, the proposal distribution that minimizes the variance of the estimator is not q(α;e) but the following.
(33)$$q_{n}^{op}\left(\alpha \right)\propto q\left(\alpha ;e\right)\left|f_{n}\left(\alpha \right)\right|,$$

That is, the optimal proposal distribution pushes the probability density to the tail of q(α;e);hence, we have reason to believe that r(α;e,τ) is closer to the optimal proposal distribution than the variational distribution q(α;e) due to the fact that, in sampling, it provides more opportunities for α, which exists in the tail of variational distribution but has higher posterior probability. Thus, (30) can be rewritten as follows:(34)$${\hat{\nabla }}_{e}ℒ=\frac{1}{M}\sum_{m}f\left(\alpha^{\left(m\right)}\right)\frac{q\left(\alpha^{\left(m\right)};e\right)}{r\left(\alpha^{\left(m\right)};e,\tau \right)},$$
where $\alpha^{\left(m\right)}$ represents the m-th sample in the sample set of r(α;e,τ).

To solve the problem of importance sampling failure caused by high-dimensional hidden variables, we use the correlation theory of mean-field, i.e., keep the rest of the components unchanged when solving the n-th component. That is to say, (34) can be expressed as follows:(35)$${\hat{\nabla }}_{e_{n}}ℒ=E_{r\left(\alpha_{n};e_{n},\tau_{n}\right)}\left[w\left(\alpha_{n}\right)E_{q\left(\alpha_{-n};e_{-n}\right)}\left[f_{n}\left(\alpha \right)\right]\right],$$
where $w\left(\alpha_{n}\right)=q\left(\alpha_{n};e_{n}\right)/r\left(\alpha_{n};e_{n},\tau_{n}\right)$, and $\alpha_{-n}$ means variables other than $\alpha_{n}$. $e_{-n}$ is the same. Inspired by the foregoing discussion, we take a simple sample $\alpha_{-n}^{0}$ from $q\left(\alpha_{-n};e_{-n}\right)$ to estimate $E_{q\left(\alpha_{-n};e_{-n}\right)}\left[\cdot \right]$ and extract M samples from $r\left(\alpha_{n};e_{n},\tau_{n}\right)$ to estimate $E_{r\left(\alpha_{n};e_{n},\tau_{n}\right)}\left[\cdot \right]$. At the same time, a score function is defined as follows.
(36)$$h_{n}\left(\alpha_{n}^{\left(m\right)}\right)=\nabla_{e_{n}}\log q\left(\alpha_{n}^{\left(m\right)};e_{n}\right).$$

It is used to calculate the following.
(37)$$f_{n}\left(\alpha^{\left(m\right)}\right)=h_{n}\left(\alpha_{n}^{\left(m\right)}\right)\log\frac{p_{n}\left(y_{k},\alpha_{n}^{\left(m\right)},\alpha_{-n}^{\left(0\right)}\right)}{q\left(\alpha_{n}^{\left(m\right)};e_{n}\right)}.$$

This formula is the embodiment of Rao–Blackwellization. Note that in order to reduce the complexity of the algorithm, we utilize $y_{k}$ instead of $Y_{k}$ in Equation (37). Moreover, $p_{n}$ is acquired by the Markov assumption. Herein, we assume that $x_{k}$ is a Gaussian distribution with ${\hat{x}}_{k}^{\alpha }$ as the mean and $P_{k}^{x,\alpha }$ as the covariance. Given the state $x_{k}$, the corresponding observation $y_{k}$ obeys the Gaussian distribution $N\left(y_{k};H_{k}x_{k},R^{\alpha_{2}}\right)$. Thus, the likelihood function of the unknown parameter α can be approximated as follows.
(38)$$f\left(y_{k}|\alpha \right)=N\left(y_{k};H_{k}x_{k},R^{\alpha_{2}}\right).$$

Consequently, we obtain the following.
(39)$$p_{n}\left(y_{k},\alpha_{n},\alpha_{-n}^{\left(0\right)}\right)=f\left(y_{k}|\alpha \right)p_{n}\left(\alpha_{n}\right)p_{-n}\left(\alpha_{-n}^{\left(0\right)}\right).$$

Finally, the gradient of ELBO relative to each component $e_{n}$ is as follows:
(40)$${\hat{\nabla }}_{e_{n}}ℒ=\frac{1}{M}\sum_{m}[f_{n}^{w}\left(\alpha_{n}^{\left(m\right)}\right)-b_{n}h_{n}^{w}\left(\alpha_{n}^{\left(m\right)}\right)]$$
where the following is the case.
(41)$$f_{n}^{w}\left(\alpha^{\left(m\right)}\right)=w\left(\alpha_{n}^{\left(m\right)}\right)f_{n}\left(\alpha^{\left(m\right)}\right),$$
(42)$$h_{n}^{w}\left(\alpha_{n}^{\left(m\right)}\right)=w\left(\alpha_{n}^{\left(m\right)}\right)h_{n}\left(\alpha_{n}^{\left(m\right)}\right),$$
(43)$$b_{n}=\frac{Cov\left(f_{n}^{w},h_{n}^{w}\right)}{var\left(h_{n}^{w}\right)}.$$

Note that $b_{n}$ marks the application of the strategy of control variates.

Next, we use the AdaGrad algorithm to make e converge to the optimal value:(44)$$e^{\left(t\right)}=e^{\left(t-1\right)}+\lambda_{t}\circ {\hat{\nabla }}_{e}ℒ,$$
where $\lambda_{t}$ is the learning rate, and its setting is similar to that in [34]. ‘∘’ stands for the Hadamard product. Note that the expectations contained in natural parameter e are the posterior noise statistics $E_{\alpha }\left[Q^{\alpha_{1}}|y_{k}\right]$ and $E_{\alpha }\left[R^{\alpha_{2}}|y_{k}\right]$ of the k-th sampling period and not the $E_{\alpha }\left[Q^{\alpha_{1}}|Y_{k}\right]$ and $E_{\alpha }\left[R^{\alpha_{2}}|Y_{k}\right]$ we require. Here, they are only needed for computing the average value from 0 to the k-th sampling period.

Furthermore, we want to automatically adjust the dispersion coefficient $\tau_{n}$ during the iteration. We first calculate the derivative of (35) with respect to $\tau_{n}$.
(45)$$-\frac{\partial V\left[{\hat{\nabla }}_{e_{n}}ℒ\right]}{\partial \tau_{n}}=\frac{1}{M}E_{r\left(\alpha_{n};e_{n},\tau_{n}\right)}\left[E_{q\left(\alpha_{-n};e_{-n}\right)}{\left[f_{n}\left(\alpha \right)\right]}^{2}w^{2}\left(\alpha_{n}\right)\frac{\partial \log r\left(\alpha_{n};e_{n},\tau_{n}\right)}{\partial \tau_{n}}\right].$$

Since $f_{n}\left(\alpha \right)$ and $w\left(\alpha_{n}\right)$ are calculated in the above process, this process will not consume much time. Next, the Monte Carlo estimation method is used to calculate the solution of the equation and then the gradient descent method is used to make t converge:(46)$$\tau_{n}^{\left(t\right)}=\tau_{n}^{\left(t-1\right)}-s_{n}\frac{\partial V\left[{\hat{\nabla }}_{e_{n}}ℒ\right]}{\partial \tau_{n}},$$
where $s_{n}$ is the step size, which needs to be set in advance. Note that if $e_{n}$ represents a vector, only the derivative part needs to be adjusted to the sum of the derivatives of all components of $e_{n}$, and Equation (46) can still be used.

Finally, if the posterior noise statistics converge to a certain value or changes little after multiple iterations, the subsequent state estimation can skip the step of finding the posterior noise parameters in order to save the algorithm overhead. The steps for implementation of the proposed KFPNS framework are summarized in Algorithms 1 and 2.
**Algorithm 1: KFPNS.**  1: **input:**
$Y_{k},p\left(\alpha \right),\Phi_{k},\Gamma_{k},H_{k},\mathrm{mean}-\mathrm{field}\;\mathrm{variational}\;\mathrm{family}\;q\left(\alpha ;e\right)$
  2: **output:**
${\hat{x}}_{k}^{\alpha }$
  3: Initialize e,τ
  4:   ${\hat{x}}_{0}^{\alpha }\leftarrow E\left[x_{0}\right]$
  5:   $E_{\alpha }\left[P_{0}^{x,\alpha }|Y_{-1}\right]\leftarrow \mathrm{initialization}$
  6:   $E_{\alpha }\left[Q^{\alpha_{1}}|Y_{-1}\right],E_{\alpha }\left[R^{\alpha_{2}}|Y_{-1}\right]\leftarrow \mathrm{Empirical}\;\mathrm{value}$
  7:   k←0
  8: **for** k=1,2,…
**do**  9: ${\tilde{z}}_{k}^{\alpha }\leftarrow y_{k}^{\alpha }-H_{k}{\hat{x}}_{k}^{\alpha }$
10: $K_{k}^{\Theta^{\prime }}\leftarrow E_{\alpha }\left[P_{k}^{x,\alpha }|Y_{k-1}\right]H_{k}^{T}E_{\alpha }{}^{-1}\left[H_{k}P_{k}^{x,\alpha }H_{k}^{T}+R^{\alpha_{2}}|Y_{k-1}\right]$
11: $E_{\alpha }\left[Q^{\alpha_{1}}|Y_{k}\right],E_{\alpha }\left[R^{\alpha_{2}}|Y_{k}\right]\leftarrow O-\mathrm{BBVI}\left(y_{k},p\left(\alpha \right)\right)$
12: $E_{\alpha }\left[P_{k+1}^{x,\alpha }|Y_{k}\right]\leftarrow \Phi_{k}\left(Ι-{Κ}_{k}^{\Theta^{\prime }}H_{k}\right)E_{\alpha }\left[P_{k}^{x,\alpha }|Y_{k}\right]\Phi_{k}^{T}+\Gamma_{k}E_{\alpha }\left[Q^{\alpha_{1}}|Y_{k}\right]\Gamma_{k}^{T}$
13: ${\hat{x}}_{k+1}^{\alpha }\leftarrow \Phi_{k}{\hat{x}}_{k}^{\alpha }+\Phi_{k}{Κ}_{k}^{\Theta }{\tilde{z}}_{k}^{\alpha }$
   **return** ${\hat{x}}_{k+1}^{\alpha }$14: **end for**


**Algorithm 2: O-BBVI.**
1: **function** O-BBVI $\left(y_{k},p\left(\alpha \right)\right)$
2: $f\left(y_{k}|\alpha \right)\leftarrow N\left(y_{k};H_{k}x_{k},R^{\alpha_{2}}\right)$
3: **While**
*the algorithm has not converged*
**do**4: Draw $\alpha^{\left(0\right)}∼q\left(\alpha ;e\right)$
5: **for** n=1 to 2
**do**6:      Draw M samples $\alpha_{n}^{\left(m\right)}∼r\left(\alpha_{n};e_{n},\tau_{n}\right)$
7:      $p_{n}\left(y_{k},\alpha_{n},\alpha_{-n}^{\left(0\right)}\right)\leftarrow f\left(y_{k}|\alpha \right)p_{n}\left(\alpha_{n}\right)p_{-n}\left(\alpha_{-n}^{\left(0\right)}\right)$
8:      **for**
m=1 to M
**do**9:   $w\left(\alpha_{n}^{\left(m\right)}\right)=q\left(\alpha_{n}^{\left(m\right)};e_{n}\right)/r\left(\alpha_{n}^{\left(m\right)};e_{n},\tau_{n}\right)$
10:   $h_{n}\left(\alpha_{n}^{\left(m\right)}\right)\leftarrow \nabla_{e_{n}}\log q\left(\alpha_{n}^{\left(m\right)};e_{n}\right)$
11:   $f_{n}\left(\alpha^{\left(m\right)}\right)\leftarrow h_{n}\left(\alpha_{n}^{\left(m\right)}\right)\log\left(p_{n}\left(y_{k},\alpha_{n}^{\left(m\right)},\alpha_{-n}^{\left(0\right)}\right)/q\left(\alpha_{n}^{\left(m\right)};e_{n}\right)\right)$
12:   $f_{n}^{w}\left(\alpha^{\left(m\right)}\right)\leftarrow w\left(\alpha_{n}^{\left(m\right)}\right)f_{n}\left(\alpha^{\left(m\right)}\right)$
13:   $h_{n}^{w}\left(\alpha_{n}^{\left(m\right)}\right)\leftarrow w\left(\alpha_{n}^{\left(m\right)}\right)h_{n}\left(\alpha_{n}^{\left(m\right)}\right)$
14:      **end for**15:      $b_{n}\leftarrow Cov\left(f_{n}^{w},h_{n}^{w}\right)/Var\left(h_{n}^{w}\right)$
16:      ${\hat{\nabla }}_{e_{n}}ℒ\leftarrow \sum_{m}[f_{n}^{w}\left(\alpha_{n}^{\left(m\right)}\right)-b_{n}h_{n}^{w}\left(\alpha_{n}^{\left(m\right)}\right)]/M$
17: **end for**18: **for** n=1 to 2
**do**19:      $-\partial V\left[{\hat{\nabla }}_{e_{n}}ℒ\right]/\partial \tau_{n}\leftarrow \mathrm{Eq}\text{.45}$
20:      $\tau_{n}\leftarrow \mathrm{Eq}\text{.46}$
21: **end for**22: set up $\lambda_{t}$
23: $e_{k}^{\left(t\right)}\leftarrow e_{k}^{\left(t-1\right)}+\lambda_{t}\circ {\hat{\nabla }}_{e}ℒ$
24: $\overset{⌢}{e}\leftarrow \mathrm{mean}\left[e_{0},\cdots ,e_{k}\right]$
25: $E_{\alpha }\left[Q^{\alpha_{1}}|Y_{k}\right]\leftarrow {\overset{⌢}{e}}_{1},\;E_{\alpha }\left[R^{\alpha_{2}}|Y_{k}\right]\leftarrow {\overset{⌢}{e}}_{2}$

**  return **

$E_{\alpha }\left[Q^{\alpha_{1}}|Y_{k}\right],\;E_{\alpha }\left[R^{\alpha_{2}}|Y_{k}\right]$



## 3. Results

In this section, we analyze the performance of the proposed algorithm and other Kalman filtering methods. The first Kalman filtering method to participate in the comparison is the IBR approach. The second one is the model-specific Kalman filtering approach, which is a classical Kalman filter designed with respect to real noise parameters. The third is the minimax method, which performs best in the worst case when the noise parameters are uncertain. We set the parameters of the minimax method to $\alpha_{\max1}=4$, $\alpha_{\max2}=4$. The last one is OBKF, which designs a message passing algorithm based on factor graphs to calculate the likelihood function formulaically and then employs the Metropolis Hastings MCMC method to find posterior effective noise statistics.

### 3.1. Simulation

For the tracking scenario in a two-dimensional space, we assume that the state of the target is $x_{k}={\left[p_{x}p_{y}v_{x}v_{y}\right]}^{T}$, and the subscripts x,y represent the x and y dimensions, respectively. Assuming that the target’s motion model is a constant velocity model and that there is only a weak random disturbance, the radar obtains the measurements of the target at every t seconds. We construct the state-space model of the measured target based on (9) and (10), where the matrices’ representations are as follows.
$$\Phi_{k}=\left[\begin{array}{cccc} 1 & 0 & t & 0\\ 0 & 1 & 0 & t\\ 0 & 0 & 1 & 0\\ 0 & 0 & 0 & 1 \end{array}\right],H_{k}=\left[\begin{array}{cccc} 1 & 0 & 0 & 0\\ 0 & 1 & 0 & 0 \end{array}\right],\Gamma_{k}=\left[\begin{array}{cc} t^{2}/2 & 0\\ 0 & t^{2}/2\\ t & 0\\ 0 & t \end{array}\right].$$

The covariance matrices of the process and observation noise are expressed as follows.
$$Q^{\alpha_{1}}=\left[\begin{array}{cc} \alpha_{1} & 0\\ 0 & \alpha_{1} \end{array}\right],R^{\alpha_{2}}=\left[\begin{array}{cc} \alpha_{2} & 0\\ 0 & \alpha_{2} \end{array}\right].$$

Let t=1 and the target’s motion state is initialized to $E[x_{0}]={\left[1035-5\right]}^{T}$ and $E_{\alpha }\left[P_{0}^{x,\alpha }|Y_{-1}\right]=diag\left(\left[1000\;1000\;100\;100\;\right]\right)$, where diag(a) represents a diagonal matrix with elements in vector a as diagonal elements.

In the first simulation, it is assumed that both $\alpha_{1}$ and $\alpha_{2}$ are unknown parameters that are uniformly distributed in the interval [1,4] and [0.5,4], respectively. In order to analyze the average performance of five Kalman filtering methods, we generate a large amount of data based on the model described above and calculate the average MSE. The results are shown in Figure 3. This set of data is generated depending on the prior distribution of noise, namely, we randomly generated 200 pairs $\left[\alpha_{1},\alpha_{2}\right]$, and each combination corresponds to 10 different observation sequences. Among the five filters, the average MSE of the model-specific Kalman filter is the smallest because it uses real noise parameters. Instead, the minimax approach only considers the worst case and cannot effectively solve the problem of unknown noise second-order statistics. Figure 3 also suggests that the average MSEs of OBKF and IBRKF are almost the same at the initial stage of filtering, and both greatly outperform KFPNS since the performance of KFPNS depends on the length of the observation sequence used to adjust filter parameters and the prior distribution of noise. If the number of sampling period k is small, the filter parameters may be adjusted incorrectly after each iteration, resulting in large state estimation errors for certain subsequent sampling periods. However, in the long run, the increase in analyzable data will return the uncertain parameters close to the true value. At this time, the performance of the KFPNS is even similar to that of OBKF. Both KFPNS and OBKF need to use a certain amount of observed data to estimate unknown noise parameters in order to achieve better estimation results, which causes them to converge more slowly than other algorithms, especially for KFPNS. The good news is that as the number of observations continues to increase, their performance is closer to the model-specific Kalman filter.

We set up two pairs of specific noise parameters for the second set of simulations. In order to discuss the average performance of various filters, we provide 200 observation sequences for each specific combination in order to weaken the interference of abnormal results. Then, we calculated the average MSE of 200 sets of observations in order to visually compare the performance of various filters. The first column of Figure 4 is a performance analysis based on the specific model $\alpha_{1}=3,\alpha_{2}=1.5$. The true value of the specific noise pair in the second column is $\alpha_{1}=3.5,\alpha_{2}=3.5$. The variation of average MSE of each filter in Figure 4a is roughly consistent with the previous simulation. Figure 4c,e, respectively, show the variation of the average posterior mean $E\left[\alpha_{1}|Y_{k}\right]$ and $E\left[\alpha_{2}|Y_{k}\right]$ under the first specific model. Similarly, (d) and (f) correspond to the second specific noise combination. The vertical columnar line represents the variance, and the size of variance is represented by the length of the line. The figures in the second and third rows show that with the increase in observation data, $E\left[\alpha_{1}|Y_{k}\right]$ and $E\left[\alpha_{2}|Y_{k}\right]$, moves closer to the true value from the preset empirical values of 2.2 and 2 and finally stabilizes near the true value. At the same time, the variance of the average posterior also means changes from small to large and then to small and eventually stabilizes. This indicates that the posterior distribution of noise estimated by KFPNS is more and more in line with the true distribution, and its value is concentrated around the true value. We also observe that in Figure 4b, the minimax method outperforms IBRKF and even outperforms OBKF in the beginning. Note that under the condition of prior knowledge, the average performance of IBRKF exceeds the minimax Kalman filter. However, IBRKF performs worse with respect to some specific models; for example, when the parameters of the minimax method are closer to the true values than the prior mean of IBRKF. Fortunately, both the OBKF and KFPNS use their approaches to approximate true posterior noise distribution, and as we observe more data, posterior second-order statistics will tend towards the true model. This allows them to break the limitations of most noise models and to deal with models that IBRKF is unable to handle.

Next, we discuss the performance changes of different robust Kalman filters when the mass of the prior probability is not uniform. Suppose $\alpha_{1}$ is uniformly distributed in the interval [1,2] and fixed. On the other hand, let a Beta distribution $ℬ(\alpha_{r},\beta_{r})$ in the interval [0.25,4] govern $\alpha_{2}$. The mean and variance of the Beta distribution are $\alpha_{r}/\alpha_{r}+\beta_{r}$ and ${\alpha_{r}\beta_{r}/\left(\alpha_{r}+\beta_{r}\right)}^{2}\left(\alpha_{r}+\beta_{r}+1\right)$, respectively, and $\alpha_{r}+\beta_{r}=1$. On the basis of ensuring that the mean of this prior distribution remains unchanged, a new prior distribution with different mass can be obtained by appropriately adjusting its parameters. We consider two pairs of specific parameters, $\alpha_{r}=0.1,\beta_{r}=0.9$ and $\alpha_{r}=0.01,\beta_{r}=0.09$. As a result, the reduction in parameters causes the variance to become larger, and the prior distribution will be more relaxed. Figure 5 shows their average MSEs. Even if the Beta distribution is changed to make the probability of values in prior distribution different, the average MSEs of the minimax method and IBRKF are almost unchanged. The reason is that for all prior distributions, the prior mean of the IBR method is always the same, while the minimax method does not consider prior distribution at all. They cannot effectively cope with sudden changes in the model. In this case, they are equivalent to the classical Kalman filter. Compared with the above two robust filtering strategies, although the mass distribution of the prior has changed, since KFPNS incorporates observations into prior knowledge in order to estimate posterior noise distribution, its average MSE is related to the tightness of the prior distribution. In other words, the tighter the prior distribution, the worse the performance of KFPNS, while the opposite is true when the prior distribution is relaxed. Importantly, the average MSE of KFPNS can always be closer to that of the model-specific Kalman filter.

In the final simulation, we analyze the performance of each robust filtering strategy when the prior knowledge is inaccurate. In practical applications, it is often impossible to understand the underlying true model, and the mastery of prior knowledge is not comprehensive. Therefore, it is crucial for a robust filter not to rely too much on the prior distribution. Our previous simulations are based on the assumption that the prior distribution of noise is accurate and available. Here, we only consider the case where the variance of the observation noise is unknown and the range of its prior distribution is included in the interval of the true distribution. We still have $\alpha_{1}$ uniformly distributed over [1,2], while assuming that $\alpha_{2}$ is uniformly distributed over [3,5], which is wrong relative to the real interval. Similarly, we generated 20 different true values based on the correct prior interval, and each true value contained 20 different sets of observations. The evaluation index of the robustness of each filter is still the average MSE. Figure 6a,b are the average MSEs of various Kalman filters in the entire sampling period when the correct interval is [2,6] and [0.5,7.5]. When k is large, the two robust filters, OBKF and KFPNS still perform best in terms of average MSEs. It is worth noting that KFPNS outperforms OBKF for the first time in Figure 6b, but it is slightly inferior or equivalent to OBKF in the previous simulation. This is related to the computing method of posterior noise statistics. When OBKF employs the Metropolis Hastings MCMC method to select sample points, the prior distribution incorporated is wrong and lacks an adjustment strategy, which makes the posterior noise statistics unable to converge to the true value. On the contrary, KFPNS adds a weighting factor in the calculation process, making it not completely dependent on the inaccurate prior distribution. Therefore, KFPNS performs better when facing models with larger errors between prior knowledge and the underlying true model. Unfortunately, the performance of KFPNS may decline when the prior distribution provided is inaccurate, and the stronger the inaccuracy, the faster the performance decline.

### 3.2. Experiment

We also processed the observed data of the real MIMO radar system. We conducted experiments in two specific scenarios. The experimental conditions are described as follows. We have delineated two fixed experimental areas, namely the indoor area of $\text{175}\times \text{465}\;{\mathrm{cm}}^{2}$ and the outdoor area of $800\times 800\;{\mathrm{cm}}^{2}$. In addition, we also calibrated some points in the two experimental areas and took these points as the ground truth of the target. Due to the limited scanning range of the MIMO radar, the position of the radar has a certain distance from the boundary of the experimental area, but we eliminated the influence of this distance in the subsequent processing. Table 2 reports the parameters of the radar used. We first sampled the environment of the two experimental areas without any targets and filtered out non-target points on this basis. During data acquisition, two people moved at a constant velocity in these two areas according to two predetermined trajectories. Figure 7 describes our experimental scenarios. Figure 7b also shows the MIMO radar board we used and the host computer interface during data acquisition. After collecting measurements, we clustered the target points and took the cluster center as the observed position of the target. Finally, four different Kalman filtering methods were used to process the data and to analyze the error.

The filters we use are the classical Kalman filter (CKF), IBRKF, OBKF and KFPNS. The filtering results under different experimental scenarios are shown in Figure 8 and Figure 9. Note that the blue point represents the estimated position of target A, and the red point represents the estimated position of target B. When filtering these position data, we do not fully grasp the underlying real model, which means that the prior knowledge we provided for these algorithms may not be accurate enough. Nevertheless, it can be observed that the estimated trajectories of the two Kalman filters designed based on posterior information are significantly smoother than the other methods. The inaccuracy of prior knowledge of noise is the main reason for the poor trajectory estimation of CKF and IBRKF. Although IBRKF introduces the concept of effective statistics, which weakens this effect to a certain extent, when the inaccuracy of prior knowledge is strong, its performance is only slightly improved compared with the classical algorithm. As expected, both KFPNS and OBKF have the characteristics of computing the posterior value and making it approximate the real value through algorithm iteration, which makes them very robust; thus, their estimated trajectory is also the closest to the true trajectory.

In addition, the position error (PE) between the estimated position and the true position will also be used as an evaluation index for filter performance. In Figure 10, although the PE of the new filter sometimes exceeds IBRKF and the classical Kalman filter, overall, most of the PE curve of KFPNS is in a lower position. Moreover, the PE of the proposed algorithm is more concentrated than compared to these two algorithms. However, in Figure 10a, it cannot be ignored that KFPNS performs poorly in the early stages of filtering. The reason for this phenomenon may be that the selection of the initial value is quite different from the real value, and there are fewer observed data available for analysis. Fortunately, with the input of more observed data, its estimation error gradually approaches OBKF.

We also compute the root-mean-square error (RMSE) of the four algorithms, and the results are shown in Figure 11. In Figure 11a, when the number of sampling period is small, the RMSE of KFPNS takes the maximum value. After 20 sampling periods, the RMSE of KFPNS tends to be stable and is significantly different from that of the other two Kalman filters with poor robustness. Figure 11b also suggests that the gap between KFPNS and OBKF is further narrowed. The change of RMSE over time proves that both KFPNS and OBKF have strong robustness. Table 3 reports the mean value of RMSE (MMSE) for various algorithms, which can be used to intuitively compare the average performance of various filters in different scenes. In the indoor scene, for target B, the average performance of KFPNS is 29.62% and 41.56% higher than that of IBRKF and the classical Kalman filter separately. Despite the average performance of KFPNS decreases by 18.1% compared with OBKF, they differ by only 0.9434 cm.

Figure 12 further analyzes the cumulative distribution function (CDF) of the RMSE of each filtering algorithm in the indoor scene. In Figure 12, 90% of the RMSE of KFPNS is less than 6.739 cm, which improves by 31% over IBRKF (9.767 cm) and 42.35% over CKF (11.69 cm) but degrades by 23.11% over OBKF (5.474 cm). In a nutshell, compared with the IBR robust filtering strategy, KNPNS and OBKF are more suitable for models where the noise is unknown and the real prior cannot be fully grasped.

### 3.3. Time Cost Analysis

Previous experiments have proved that OBKF and KFPNS have considerable robustness. Now we study the difference of their complexity. Since the recursive structure of these two robust Kalman filters is consistent with the classical algorithm, their computational burden is mainly concentrated on the calculation of posterior noise statistics. It is worth mentioning that OBKF uses a factor-graph-based method to convert the problem of finding the likelihood function into a matrix operation, which increases the complexity of the algorithm. Moreover, if the posterior noise statistics at the k-th observation need to be computed, the likelihood function of each MCMC sample must be iterated from i=0 until i=k-1. Therefore, its computational complexity depends on the dimension of the matrix, the number of samples in the posterior distribution and the number of sampling periods. In contrast, the proposed algorithm rarely involves matrix operations, and there is no need to compute the likelihood function from the beginning. At the same time, the variational distribution is fixedly decomposed into two factors. These reasons all reduce the calculation burden of KFPNS to a considerable extent. Figure 13 shows the average run times of the OBKF and KFPNS relative to the sampling period. Note that both algorithms need to generate 10,000 samples to approximate the true posterior noise distribution. In addition, the running platform of all simulations and experiments in this paper is MATLAB R2018b with Intel(R) Core(TM) i5-8257U CPU at 1.40 GHz as well as 8-GB RAM. Figure 13 illustrates that under the same number of samples and the same length of observation sequence, OBKF needs to consume more time. Consequently, KFPNS performs better in real time and is more suitable for applications in actual projects.

## 4. Discussion

In this paper, the IBR framework is extended to a posteriori version by using posterior effective noise information. More exactly, a Bayesian robust Kalman filter based on posterior noise statistics is designed by adding the calculation method of posterior noise statistics. Our simulation results also suggest that when dealing with models with uncertain noise parameters, the performance of the designed filter is second only to OBKF. Therefore, it can be concluded that the proposed calculation method of posterior noise statistics can make the uncertain noise parameters converge near the true value. When used in an actual MIMO radar system, we found that our algorithm has a good performance in robustness and real-time situations compared with current methods. However, it is worth noting that since the run time of the KFPNS is proportional to the length of the observation sequence, when the tracking period is longer, it will consume more time than compared to the classic algorithm. Furthermore, our algorithm is run under the assumption of a Gaussian model. In other words, if the noise is non-Gaussian, the performance of our algorithm will be greatly reduced. In future investigations, we will propose a more effective and faster method for estimating posterior noise distribution and make more classes of noise suitable for this framework.

## Figures and Tables

**Figure 1 sensors-21-07673-f001:**
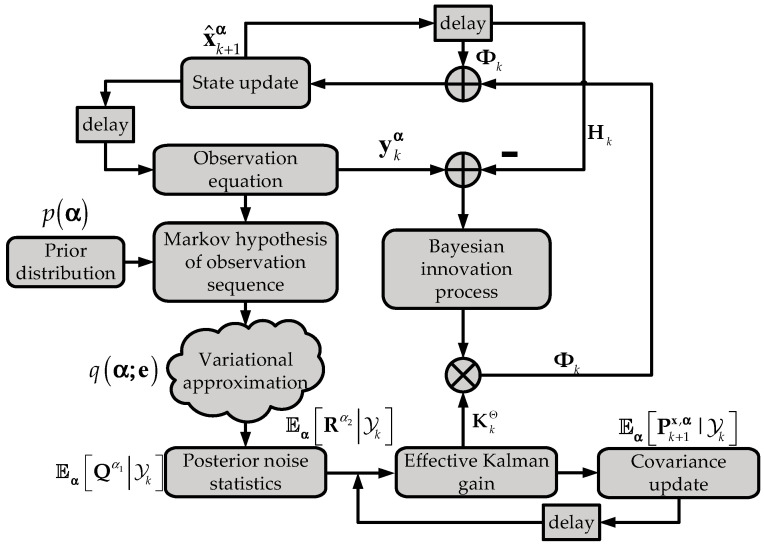
KFPNS framework.

**Figure 2 sensors-21-07673-f002:**
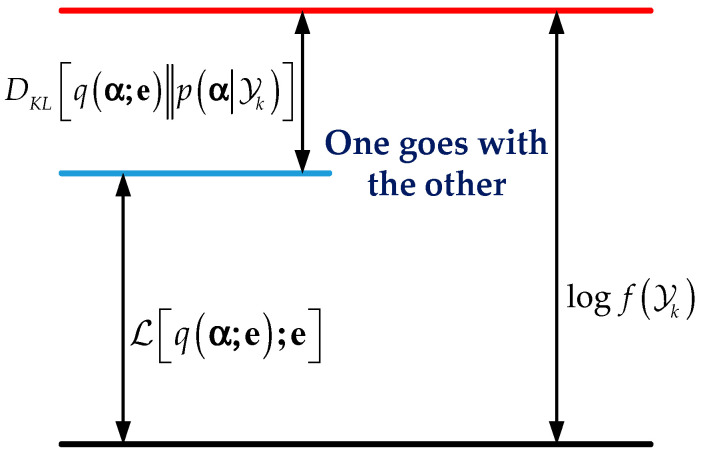
The principle of conversion.

**Figure 3 sensors-21-07673-f003:**
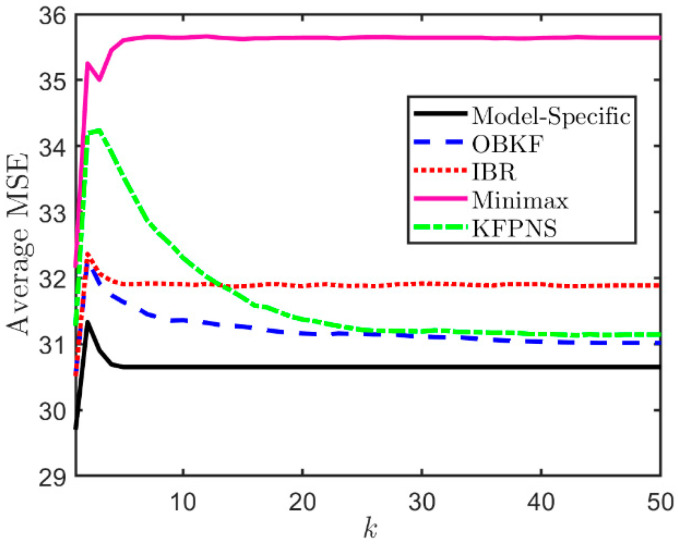
Average MSE of various filters when $\alpha_{1}$ and $\alpha_{2}$ are uncertain.

**Figure 4 sensors-21-07673-f004:**
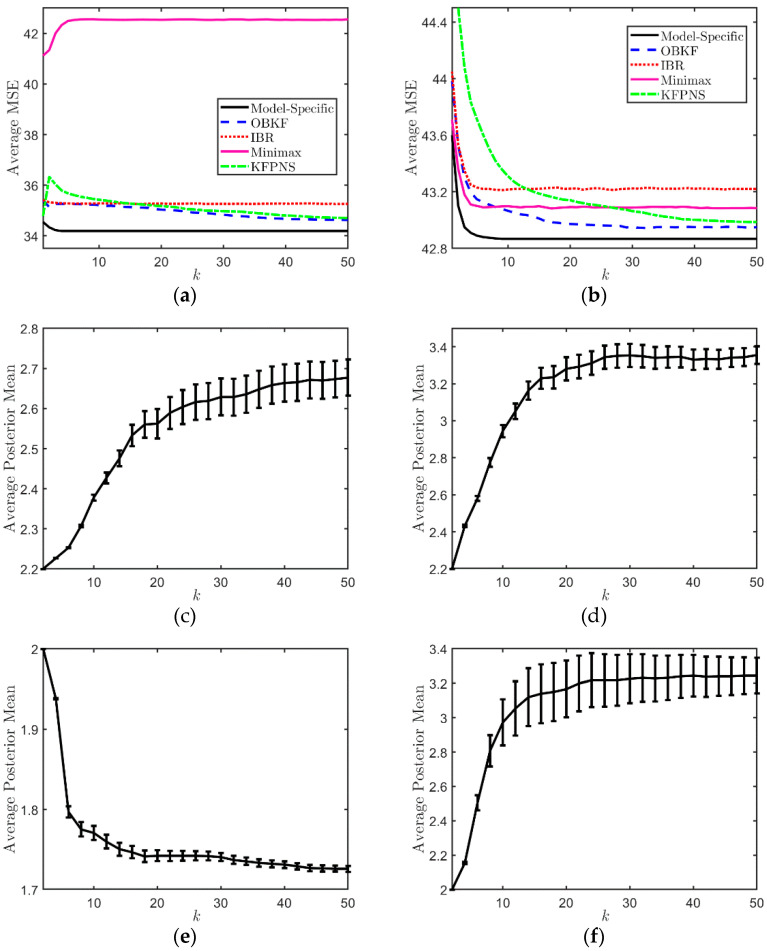
Performance analysis for specific noise pairs. (**a**) The average MSE for specific noise model $\alpha_{1}=3,\alpha_{2}=1.5$. (**b**) The average MSE for specific noise model $\alpha_{1}=3.5,\alpha_{2}=3.5$. (**c**) The variation of $E\left[\alpha_{1}|Y_{k}\right]$ and its variance when $\alpha_{1}=3$. (**d**) The variation of $E\left[\alpha_{1}|Y_{k}\right]$ and its variance when $\alpha_{1}=3.5$. (**e**) The variation of $E\left[\alpha_{2}|Y_{k}\right]$ and its variance when $\alpha_{2}=1.5$. (**f**) The variation of $E\left[\alpha_{2}|Y_{k}\right]$ and its variance when $\alpha_{2}=3.5$.

**Figure 5 sensors-21-07673-f005:**
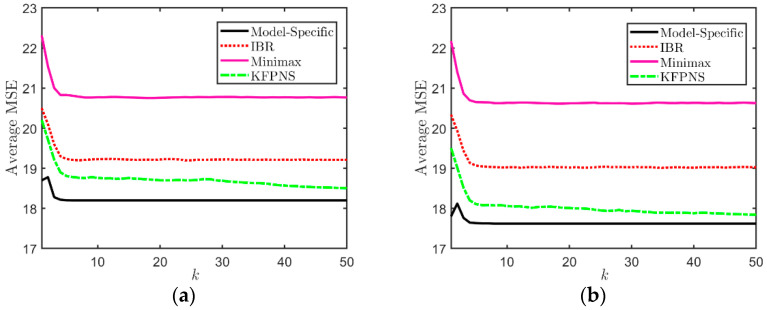
The average performance of four filters with different Beta priors. (**a**) $\alpha_{r}=0.1,\beta_{r}=0.9$; (**b**) $\alpha_{r}=0.01,\beta_{r}=0.09$.

**Figure 6 sensors-21-07673-f006:**
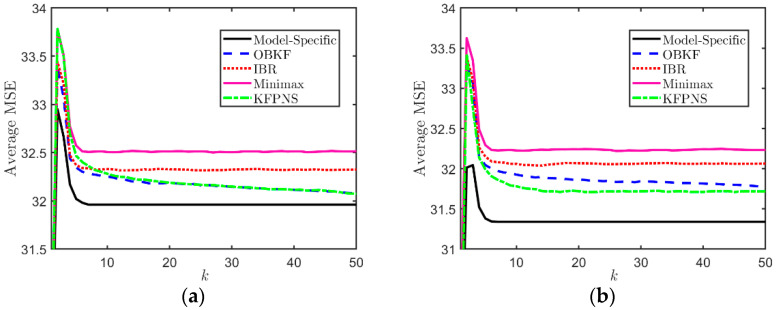
Average MSE of various filters when $\alpha_{2}$ is uncertain, and the interval [3,5] of prior distribution is inaccurate. (**a**) The exact interval is [2,6]. (**b**) The exact interval is [0.5,7.5].

**Figure 7 sensors-21-07673-f007:**
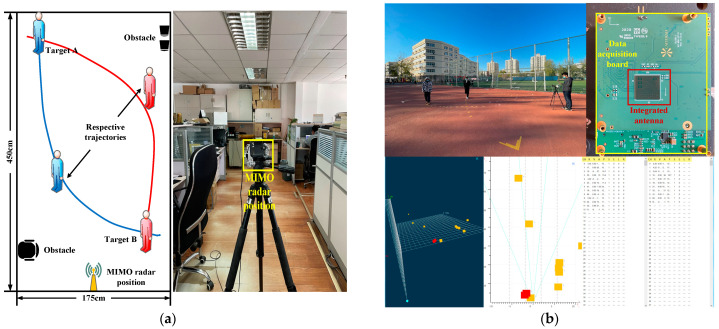
Real data acquisition scenes. (**a**) Indoor scene. (**b**) Outdoor scene.

**Figure 8 sensors-21-07673-f008:**
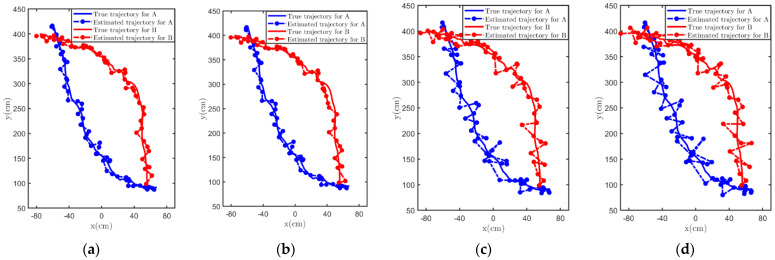
The fitting degree between the estimated trajectory and the true trajectory after processing by various Kalman filtering methods in an indoor scene. (**a**) OBKF. (**b**) KFPNS. (**c**) IBRKF. (**d**) CKF.

**Figure 9 sensors-21-07673-f009:**
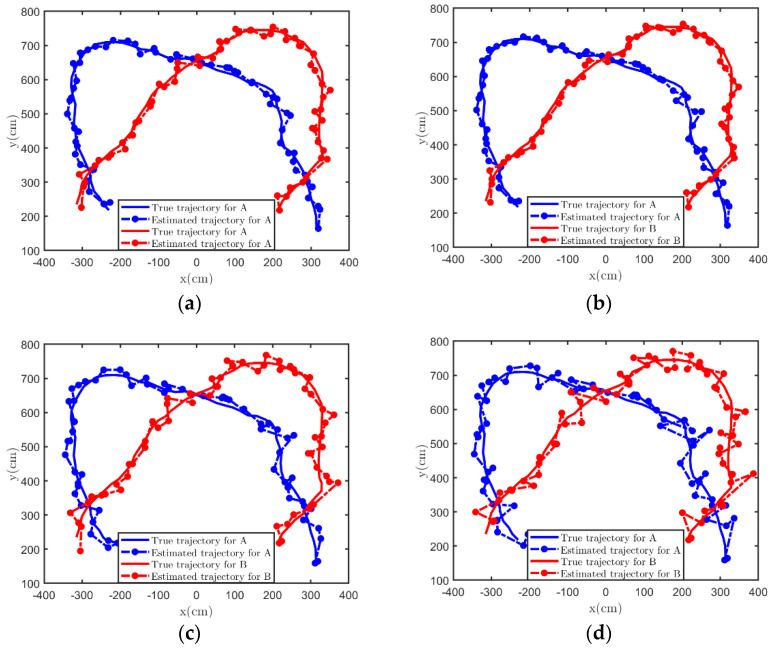
The fitting degree between the estimated trajectory and the true trajectory after processing by various Kalman filtering methods in an outdoor scene. (**a**) OBKF. (**b**) KFPNS. (**c**) IBRKF. (**d**) CKF.

**Figure 10 sensors-21-07673-f010:**
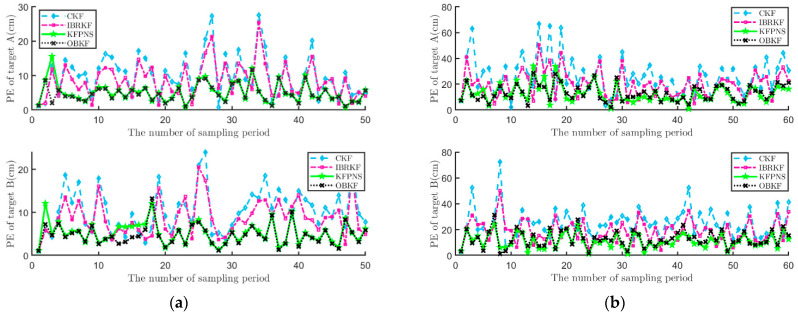
The PE of each filter in two specific scenes. (**a**) Comparison of PE of various Kalman filters in an indoor scene. (**b**) Comparison of PE of various Kalman filters in an outdoor scene.

**Figure 11 sensors-21-07673-f011:**
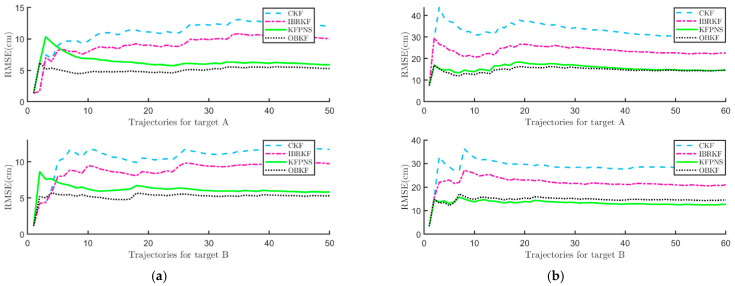
The RMSE of various Kalman filtering methods in different scenes. (**a**) The RMSE of the four Kalman filtering methods in an indoor scene with respect to target A and target B, respectively. (**b**) The RMSE of the four Kalman filtering methods in an outdoor scene with respect to target A and target B, respectively.

**Figure 12 sensors-21-07673-f012:**
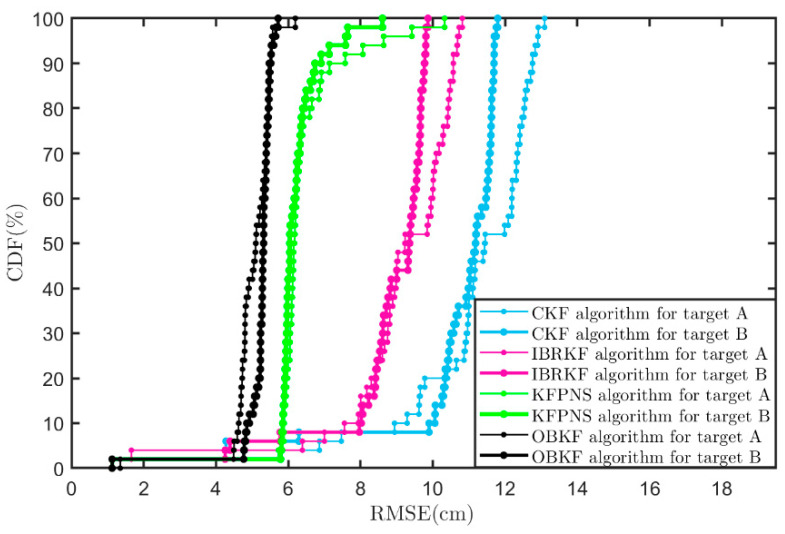
CDFs of RMSE for various algorithms in an indoor scene.

**Figure 13 sensors-21-07673-f013:**
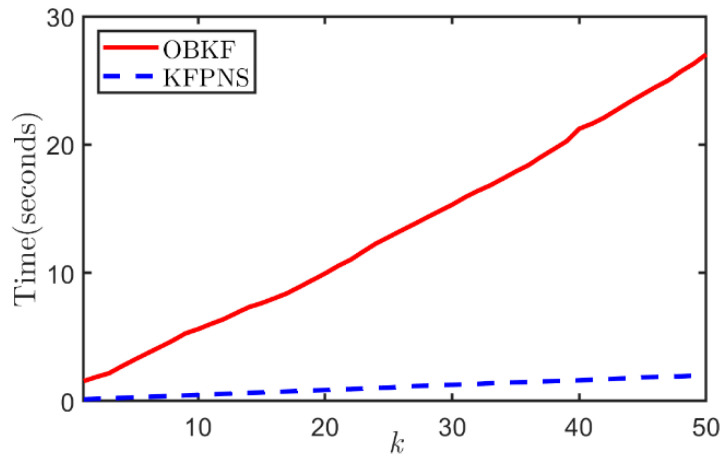
The time consumption of two algorithms when the length of the observation sequence changes.

**Table 1 sensors-21-07673-t001:** Comparison of the recursive equations for the classical algorithm and KFPNS.

Classical Kalman Filter	KFPNS
${\tilde{z}}_{k}=y_{k}-H_{k}{\hat{x}}_{k}$	${\tilde{z}}_{k}^{\alpha }=y_{k}^{\alpha }-H_{k}{\hat{x}}_{k}^{\alpha }$
$K_{k}=P_{k}^{x}H_{k}^{T}{\left(H_{k}P_{k}^{x}H_{k}^{T}+R\right)}^{-1}$	$K_{k}^{\Theta^{\prime }}=E_{\alpha }\left[P_{k}^{x,\alpha }|Y_{k-1}\right]H_{k}^{T}E_{\alpha }{}^{-1}\left[H_{k}P_{k}^{x,\alpha }H_{k}^{T}+R^{\alpha_{2}}|Y_{k-1}\right]$
${\hat{x}}_{k+1}=\Phi_{k}{\hat{x}}_{k}+\Phi_{k}{Κ}_{k}{\tilde{z}}_{k}$	${\hat{x}}_{k+1}^{\alpha }=\Phi_{k}{\hat{x}}_{k}^{\alpha }+\Phi_{k}{Κ}_{k}^{\Theta }{\tilde{z}}_{k}^{\alpha }$
$P_{k+1}^{x}=\Phi_{k}\left(Ι-{Κ}_{k}H_{k}\right)P_{k}^{x}\Phi_{k}^{T}+\Gamma_{k}Q\Gamma_{k}^{T}$	$\begin{array}{c} E_{\alpha }\left[P_{k+1}^{x,\alpha }|Y_{k}\right]=\Phi_{k}\left(Ι-{Κ}_{k}^{\Theta^{\prime }}H_{k}\right)E_{\alpha }\left[P_{k}^{x,\alpha }|Y_{k}\right]\Phi_{k}^{T}\\ +\Gamma_{k}E_{\alpha }\left[Q^{\alpha_{1}}|Y_{k}\right]\Gamma_{k}^{T} \end{array}$

**Table 2 sensors-21-07673-t002:** MIMO radar parameters.

Radar Model	Radar System	The Start Frequency	Range Resolution	Frame Periodicity	Scan Range
RDP-77S244-ABM-AIP	FMCW	77 GHz	0.045 m	200 ms	FOV 120°

**Table 3 sensors-21-07673-t003:** The MMSE difference of various Kalman filtering methods.

Algorithm	MMSE			
	Indoor Scene		Outdoor Scene	
	Target A	Target B	Target A	Target B
KFPNS	6.3679 cm	6.1552 cm	15.5520 cm	13.1918 cm
IBRKF	9.0658 cm	8.7459 cm	23.5724 cm	21.7488 cm
CKF	11.0092 cm	10.5480 cm	32.6647 cm	28.4253 cm
OBKF	5.0358 cm	5.2118 cm	14.6269 cm	14.6148 cm

## Data Availability

Not applicable.

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
