# Peer review of "An Overdispersed Black-Box Variational Bayesian–Kalman Filter with Inaccurate Noise Second-Order Statistics"

_sensors, 2021, doi:10.3390/s21227673_

Round 1

Reviewer 1 Report

To solve the state estimation problem with unknown noise statistics, this paper proposes a robust Kalman filter under the framework of over dispersed black-box variational inference. This idea is interesting. However, there are some confusing points in this paper.

  1. In the introduction, the authors mainly introduce only one method (IBRKF)to solve similar problems. As far as I know, there are still many related references that solve similar problems. The author should supplement these references and add corresponding comparative experiments in the simulation as much as possible.
  2. The unclear expression of related variables (e.g. the natural parameters, the sufficient statistics) makes it difficult to understand the principle of the algorithm.
  3. The author should explain why the variational distribution of α needs to belong to the exponential family.
  4. The author should explain the difference between the over dispersed black box variational inference and the black box variational Bayesian.

Reviewer 2 Report

Title does not make any reference to the framework of application. More detail is needed.

First phrase of the Abstract is too straightforward. More approximation to the context is needed from the beginning.

More applications of Kalman for trajectory/localization of moving agents are expected in the state-of-the-art section.

More experimental data is expected for such kind of work. The real scenario proposed is too small and custom. What is the ground truth calculated, and so that the estimation error? Why not testing the framework in a larger and/or outdoors scenario? What about the acquisition system?

It is crucial the previous questions are addressed and more experimental setup and data provided

Round 2

Reviewer 1 Report

My comments have been addressed sufficiently. This manuscript can be published.

Reviewer 2 Report

My comments have been addressed sufficiently.

This manuscript is a resubmission of an earlier submission. The following is a list of the peer review reports and author responses from that submission.